# Methods

# A FIJI macro for quantifying pattern in extracellular matrix

Esther Wershof[1,2,3], Danielle Park[4,*], David J Barry[5,*] , Robert P Jenkins[2], Antonio Rullan[2] , Anna Wilkins[2],
Karin Schlegelmilch[2], Ioannis Roxanis[6], Kurt I Anderson[5], Paul A Bates[1] , Erik Sahai[2]

**Diverse extracellular matrix patterns are observed in both normal and pathological tissue. However, most current tools for quantitative analysis focus on a single aspect of matrix patterning. Thus, an automated pipeline that simultaneously quantifies a broad range of metrics and enables a comprehensive description of varied matrix patterns is needed. To this end, we have developed an ImageJ plugin called TWOMBLI, which stands for The Workflow Of Matrix BioLogy Informatics. This pipeline includes metrics of matrix alignment, length, branching, end points, gaps, fractal dimension, curvature, and the distribution of fibre thickness. TWOMBLI is designed to be quick, versatile and easy-to-use particularly for non-computational scientists. TWOMBLI can be downloaded from https://github.com/wershofe/TWOMBLI together with detailed documentation and tutorial video. Although developed with the extracellular matrix in mind, TWOMBLI is versatile and can be applied to vascular and cytoskeletal networks. Here we present an overview of the pipeline together with examples from a wide range of contexts where matrix patterns are generated.**

## Introduction

The ECM provides support and structure to multicellular organisms and also guides the migration of cells (1, 2), including leukocytes engaged in immune surveillance (3). Changes in the ECM are central to the ageing process, the ECM is re-built and remodelled in response to tissue damage, and it is further altered in pathologies such as cancer. The architecture of the ECM has lately been the subject of renewed focus (4, 5, 6); however, standardised quantification of ECM patterns, particularly in pathological decision-making is lacking. The generation of metrics that describe ECM pattern could lead to insights into a wide range of fields, ranging from experimentalists interested in cell migration and remodelling of matrices to clinicians researching conditions such as cancer and fibrosis.

Diverse ECM patterns are observed in both normal and pathological tissue. Highly aligned and linear matrix fibres are well suited to resisting tensile stresses in the direction of alignment, and can also serve as migration routes for cells. Furthermore, the transition of isotropic and curved matrices into more linear aligned matrices is associated with aggressive cancers. Numerous metrics have already been applied to such ECM images. These range from simple abundance and area measurements to more complex textural features such as grey level co-occurrence matrices (7), which measures texture by analysing the variation in pixel intensity value, and box-counting fractal dimension (8), which measures how an image is sampled in "boxes" of decreasing size. One commonly used metric is matrix fibre alignment, which is known to be a promoter of cancer cell invasion (9, 10, 11). A pipeline already exists in MATLAB for quantifying alignment of ECM (12), and a further extension of this work enables alignment to be related to the tumour margin (13). OrientationJ is an automated ImageJ plugin that is able to create vector fields and perform directional analysis on fibres, but this does not lend itself to quantifying overall matrix patterns. A number of studies have attempted to quantify additional matrix metrics, but these typically require heavy manual intervention (14, 15). Furthermore, each tool typically only generates metrics relating to one aspect of ECM organisation. This makes collating a broad range of metrics for analysis both challenging and time-consuming. There is a need for an end-to-end pipeline for quantifying ECM patterns, which is automated and easy-to-use on versatile data sets. To this end, we have created the ImageJ macro plugin TWOMBLI, which stands for The Workflow Of Matrix BioLogy Informatics. The aim of TWOMBLI is to quantify matrix patterns in an ECM image by deriving a range of metrics, which can then be analysed in conjunction with clinical data if desired.

## Results

We sought to generate a tool for analysis of ECM patterns, whereby a user could enter varied matrix images into the pipeline and derive a meaningful quantification of a variety of matrix features as output (Fig 1). In particular, we were keen that

[1]Biomolecular Modelling Laboratory, The Francis Crick Institute, London, UK    [2]Tumour Cell Biology Laboratory, The Francis Crick Institute, London, UK    [3]Sloan Kettering Institute, Memorial Sloan Kettering Cancer Center, New York City, NY, USA    [4]Developmental Signalling Laboratory, The Francis Crick Institute, London, UK    [5]Advanced Light Microscopy Facility, The Francis Crick Institute, London, UK    [6]Breast Cancer Research Division, Toby Robins Breast Cancer Now Research Centre, The Institute of Cancer Research, London, UK

Correspondence: paul.bates@crick.ac.uk; erik.sahai@crick.ac.uk
*Danielle Park and David J Barry contributed equally to this work

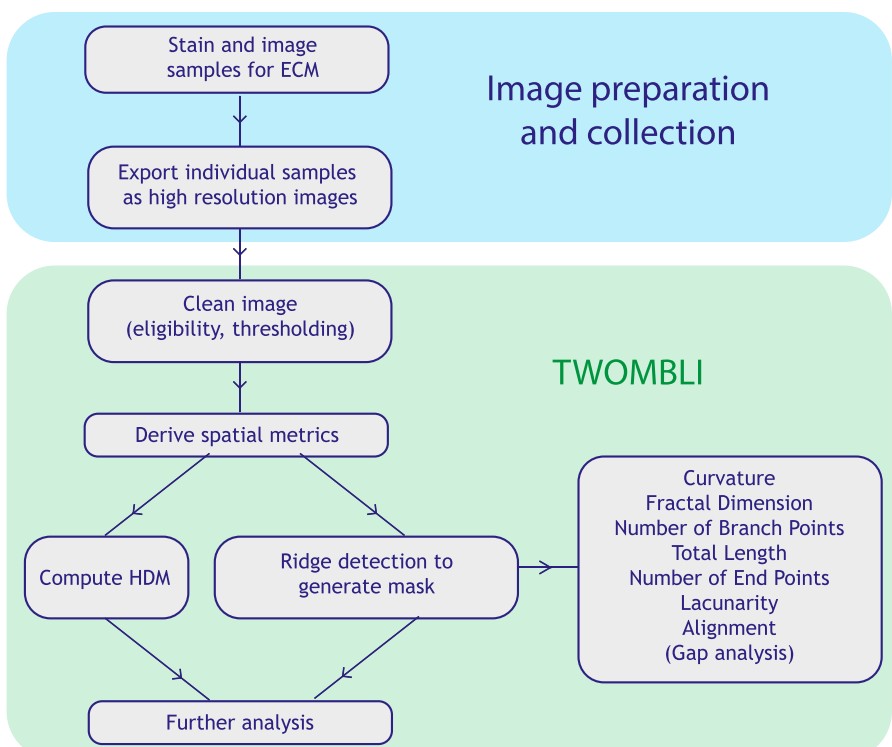

**Figure 1. Workflow diagram of quantification of matrix patterns.**
End-to-end pipeline from obtaining the samples through matrix quantification to survival analysis based on this matrix metrology. A list of metrics is given in the right-most box.

a broad range of metrics would capture diverse features of ECM patterns. FIJI was used as the supporting platform for the plugin because it enabled us to build on many existing tools and downloads such as Ridge Detection (16), Anamorf (17), OrientationJ, and BIOP and is also familiar to many scientists and clinicians.

### Development of the TWOMBLI pipeline

TWOMBLI exists as a user-friendly ImageJ plugin that can be downloaded with detailed documentation from https://github.com/wershofe/TWOMBLI. TWOMBLI relies heavily on existing ImageJ plugins Ridge Detection and AnaMorf (16, 17), providing an end-to-end pipeline for users. Ridge detection is an algorithm for detecting the filamentous structure of an image in an unbiased manner and Anamorf is a plugin for quantifying such structures. TWOMBLI uses some of these metrics and adds others more pertinent to the study of matrix fibres. For example, the dominant direction generated by OrientationJ provides a global alignment metric, which is not provided by Anamorf but is important to know in the context of fibre organisation. The user can input tissue samples stained for ECM components and is then guided through image pre-processing before finally being given a comprehensive output of ECM metrics in a single csv file with accompanying processed images. Central to this process is the generation of mask files of the matrix network using the Ridge Detection tool. Matrix fibres, which can be considered as ridges in the image, are detected using spatial derivatives, which report the rate of change in intensity signal as a function of distance in nearby pixels. The edges of fibres have a high local change in intensity value and this

information is used to construct lines, making it computationally efficient. The algorithm also allows for subpixel detection of lines and is robust to an asymmetric contrast gradient on either lateral edge of the line allowing for good line extraction even in low resolution images. The mask files are generated as outputs of the pipeline to enable visual verification of correct thresholding. Furthermore, an additional script is included in the TWOMBLI repository for optional cropping down of images to regions of interest (ROIs). The principal stages of TWOMBLI are prechecking, preprocessing, and processing. The prechecks (steps 0–3) are carried out manually by the user to check the eligibility of potential input images following prompts. An image needs to be in focus, have high enough resolution, and not contain too many artefacts or regions that are not pertinent for the analysis. In pre-processing (steps 4–10), the user is guided through selecting a subset of test images and then choosing appropriate parameters for thresholding these test images. In addition to contrast saturation, which relates obtaining the appropriate contrast between areas of matrix and no matrix for subsequent steps, a line width parameter is requested. This value sets the thickness of matrix fibres that will be identified. TWOMBLI also has the capability to record fibres in a range of thicknesses, which the user can specify. This can be useful in the event that an image contains both thicker and thinner fibres. These parameters are saved so that in future runs, the user can skip directly to the processing stage. In processing (steps 11–14), all of the images are analysed in batch using the parameters generated in the pre-processing stage. An input image of 1 MB in size takes ~30 s to process on a computer with 2.2 GHz Intel Core i7 and 16 GB of memory. For tips on handling larger files and other troubleshooting, see Documentation (Supplemental Data 1).

**Life Science Alliance**

## Metrics for matrix quantification

The diversity of ECM organisation is highlighted in Fig 2 using examples of breast cancer biopsies. We selected metrics designed to capture the different aspects of ECM patterning. Fig 2 shows schematic illustrations of the metrics are next to the relevant breast cancer images. These metrics can be split between those describing individual fibres and those describing general ECM patterning: number of fibre end points, number of fibre branch points, total length of fibres, fibre curvature, alignment, proportion of high-density matrix (HDM), ECM fractal dimension, hyphal growth unit (HGU) (a measure of the number of end points per unit length), matrix gaps, and lacunarity (a measure of how the ECM fills the space) (17). These metrics together could characterise different matrix properties. Apart from HDM, all metrics are calculated from the mask images generated within the TWOMBLI pipeline.

### *Quantification of individual fibres*

Number of end points is an intuitive count of the number of ends of the fibres or filaments in the mask image. Number of branch points is the number of intersections of mask fibres in the image. Total length is the sum of the length of all mask fibres in the image. This quantity can be useful in normalising the number of branch points and end points. The HGU corresponds to the number of end points per unit length. Furthermore, the average fibre length is easily calculated by dividing the total fibre length by 0.5(E + B), where E = end points and B = branch points and it is assumed that one of the fibres involved at a branch begins at the branch point. An estimate of fibre thickness can be derived by dividing the HDM value (matrix area) by the fibre length. Curvature is measured as the mean change in angle moving incrementally along individual mask fibres by user-specified windows. TWOMBLI has the capability to record curvature for a range of curvature windows.

### *Quantification of global pattern*

HDM is a measure of the proportion of pixels in an image corresponding to matrix, as defined by the user-specified contrast saturation parameter and subsequent thresholding. The alignment metric captures the extent to which fibres within the field of view are oriented in a similar direction (15, 18). It is calculated from the global gradient structure tensor defined as follows:

$$J = \left[ \langle f_x, f_x \rangle_w \langle f_x, f_y \rangle_w \langle f_x, f_y \rangle_w \langle f_y, f_y \rangle_w \right],$$

where $f_x$ and $f_y$ are partial spatial derivatives of the image, $f(x, y)$, and

$$\langle g, h \rangle_w = \iint_{R^2} w(x, y) \, g(x, y) \, h(x, y) \, dxdy.$$

The function, $w(x, y)$ is a normalised weighting window centred on the ROI.

Alignment is then determined from the coherency metric

$$a = \frac{\lambda_{max} - \lambda_{min}}{\lambda_{max} + \lambda_{min}},$$

where $\lambda_{min}$ is the smallest eigenvalue (minor axis) and $\lambda_{max}$ is the largest eigenvalue (major axis) of the tensor, yielding a value in the range [0, 1], where zero represents complete isotropy and one represents perfect alignment. Fractal dimension is an indicator of the self-similarity and complexity of the ECM and is bound between the range [1, 2] for a single 2D image slice. Specifically, the metric used is the box-counting dimension (19). A grid with squares of side length $\epsilon$ is overlaid over the image. The number of squares $N(\epsilon)$ which are occupied by the non-background part of the image N is recorded. As $\epsilon$ gets smaller, $N(\epsilon)$ increases. Fractal dimension is then computed as the limit of the following equation:

$$frac = \frac{logN(\epsilon)}{log(1/\epsilon)}.$$

The lacunarity metric reflects the number and size of gaps in the matrix. Briefly, the variation in pixel intensity is sampled in different directions and in different size grids and a single average value is returned, with larger values indicating larger space in the matrix pattern. Formally, lacunarity is quantified as follows:

$$\Lambda = \left| \frac{s^2}{\mu^2} - 1 \right|,$$

where $\mu$ and $s$ are, respectively, the mean and standard deviation of grey level enclosed within a region (20). In addition, an optional step in the plugin allows the user to perform gap analysis on matrix patterns using the Max Inscribed Circle function available from the BIOP plugin in FIJI. The algorithm analyses spaces between objects (in this case, the fibres in the masks derived by TWOMBLI) by fitting circles of decreasing radius to fill in the gaps (21). This information is reported in individual .csv files containing the size of all gaps identified, which allows for individual researchers to choose whether to focus the average size of gaps, the shape of the distribution, or even the size of gaps in the tails of the distribution. Depending on the tissue and context, these different metrics of gap sizes provide additional insight into the structure of the matrix patterns.

## Use of the TWOMBLI pipeline

A tutorial on using TWOMBLI can be found in https://github.com/wershofe/TWOMBLI. Fig 3A shows an example of user inputs together with the different outputs from the TWOMBLI pipeline (images of human breast cancer from Fig 2 are shown for illustrative purposes). The example shown involves Picrosirius red staining of fibrillar collagen in a breast cancer biopsy, with the matrix filaments identified shown in the middle image. Importantly, the pipeline can be used to analyse images of ECM acquired via a wide variety of imaging techniques and of varying file types. The user is prompted to input whether the fibres are light on a dark background (as in fluorescence imaging) or dark on a light background (as in conventional histological imaging). Accurate filament identification

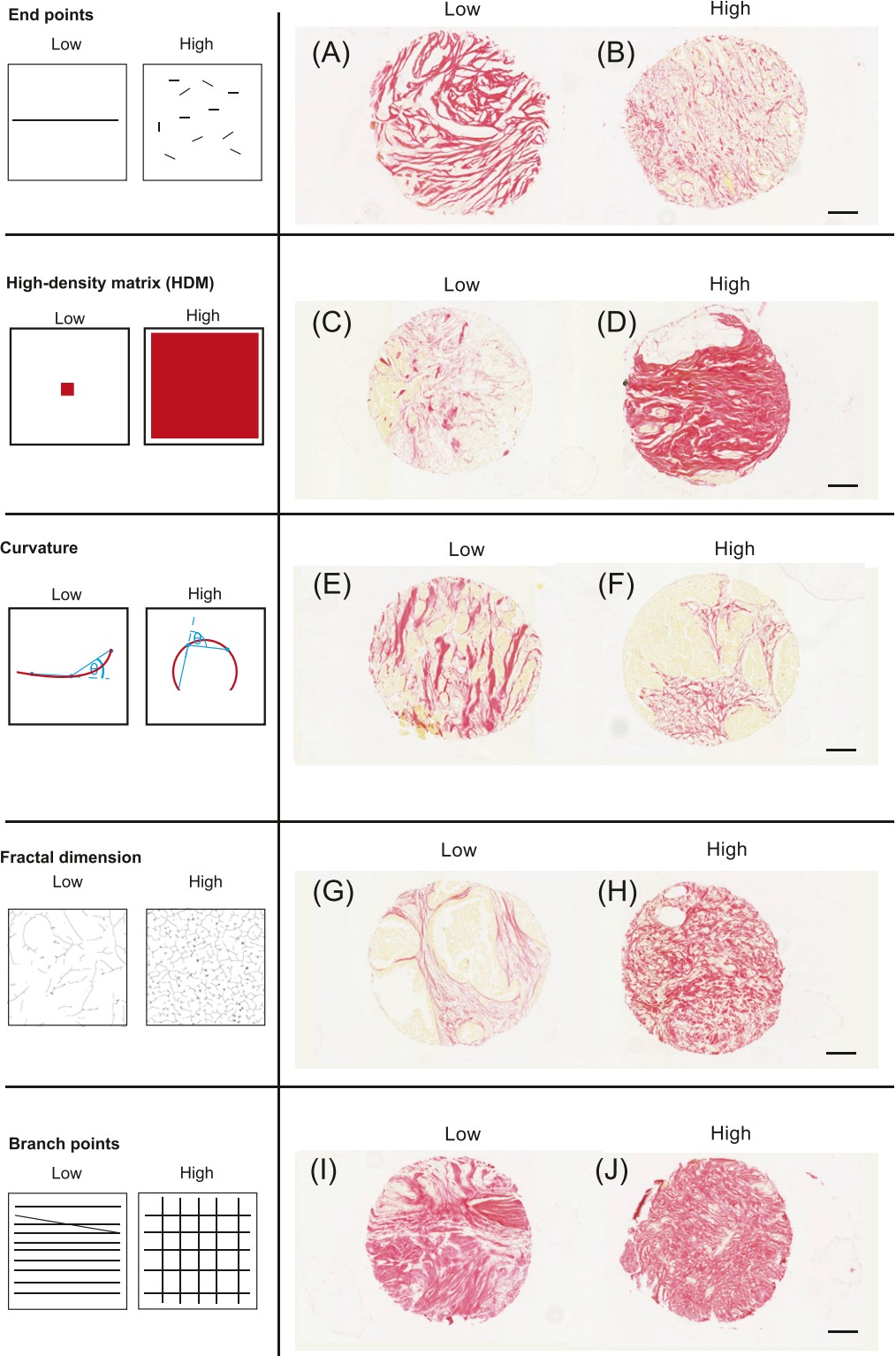

**Figure 2. Schematics and example tissue biopsies of ECM metrics.**
**(A, B, C, D, E, F, G, H, I, J)** Images show Picrosirius red staining breast cancer biopsies. Each biopsy is 600 $\mu$m in diameter, scale bar is 100 $\mu$m.

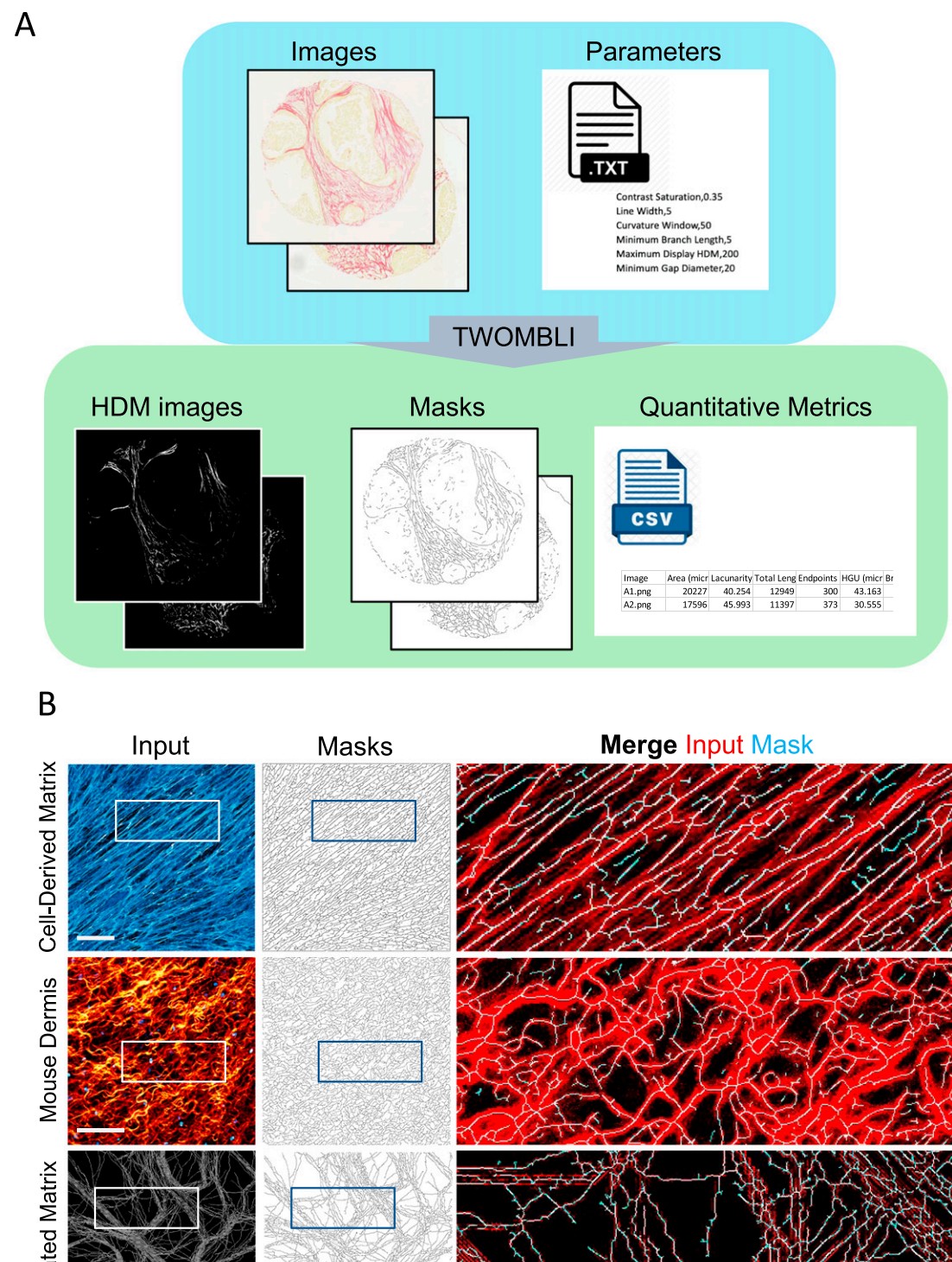

**Figure 3.   Input and outputs of TWOMBLI pipeline.**
**(A)** User inputs an image of a sample stained for ECM components (in this case, collagen is stained with Picrosirius red). Outputs consist of a mask, a csv file containing matrix metrics based on the mask and a processed image thresholded for HDM. Images from Fig 2 are used for illustrative purposes to demonstrate the quantification of images that look qualitatively different. **(B)** Example input images can be acquired from a range of sources: for example (from left to right), fibronectin-stained CDM in vitro (images are 500 × 500 μm, fibronectin in blue), second-harmonic imaging in vivo (images are 400 × 400 μm, collagen in orange and fibroblast nuclei in blue), and in silico ECM. Magnified panels show input matrix in red and mask in cyan. Scale bars are 100 μm.

also depends on the "Contrast Saturation" parameter and users are advised to check that both the HDM and mask image outputs correctly reflect the input data. Fig 3B shows its application to cell-derived matrix (CDM) assays (isotropic), second-harmonic imaging of mouse tissues, and synthetic matrix patterns generated via computer simulations (5). In all examples, the algorithm is able to correctly identify matrix filaments, which are shown in the Mask images (Fig 3B). For histochemical staining methods that include a counterstain with a different colour from that of the matrix stain, we recommend using the colour deconvolution tool in FIJI before running the images in TWOMBLI. Fig S1 shows examples of this applied to both Picrosirous red staining with a haematoxylin counterstain and Masson's trichrome. For best results, we advise bespoke definition of the colour deconvolution matrix. Fig S1 shows how this works based on the selection of three different ROIs.

In addition to the image inputs, a parameter file is required (see Supplemental Data 1 page 12 and Tutorial Video (https://github.com/wershofe/TWOMBLI) minutes 15:30–16:28). The contrast saturation and HDM values relate to the thresholding of the input images. The minimum branch size and minimum gap size, specified in pixel units, are intended to avoid erroneous identification of very small filaments. Suggested starting parameters are provided in the parameter file that can be downloaded along with the tutorial images. However, empirical refinement is advised using a subset of the images to be analysed. Line width and curvature window parameters should be informed by the dominant filament size and curvature scale in the images. Fig S2 shows that larger curvature windows preferentially pick up curved filaments with a larger length scale to their waviness or larger curve radius (compare example filaments "i" and "ii" in Fig S2A). The tighter curvature of the spleen scores higher than the mammary fat pad if a curvature window of 10 is used, but lower if larger curvature windows are used (Fig S2B). In all cases, the linear fibres of mammary tumour matrix generate a low score. If users are interested in curvature over different length scales then we advise running analysis at multiple curvature windows (as reported in Fig S2). This will lead to a more complex dataset, but it will be correspondingly richer. Larger line width values will limit TWOMBLI to the detection of larger fibres. TWOMBLI prompts the user for a maximum and minimum line width. If the same values are entered, then only a single value line width will be evaluated. If different line widths are entered, then TWOMBLI will analyse intermediate line widths in increments of five. The same input images as used in Fig S2 are then used in Fig S3, showing how varying line width influences the masks that are generated and how the same images are handled by different aspects of TWOMBLI. If images have a wide array of matrix fibre thicknesses, then it is possible to generate a mask that amalgamates the masks generated using different line widths – although it comes at the cost of computational time. Examples of this are shown in Fig S3 (lower panels).

The utility of quantification tools is crucially dependent upon their robustness to variations in the exact details of data acquisition. To explore this, we captured images of regions of two CDMs with different features: one was largely isotropic with finer fibres (hereafter referred to as isotropic, Fig S4A) and the other more anisotropic with thicker, sparser, fibres (hereafter referred to as anisotropic, Fig S4A). Images of these two matrices were acquired using different microscope objectives, gain settings, pixel sizes, and focal planes (either different focal planes of a CDM captured using single confocal sections with a narrow pinhole, a single confocal section with a wide pinhole setting, or a z projection of a stack of confocal sections). These images were then analysed using two different line width settings in TWOMBLI. In addition, we electronically degraded the quality of some images using Gaussian blur functions of either two or three pixels radius or the standard noise function in ImageJ. This set of images were then run through TWOMBLI. Table S1 shows the output metrics and Fig S4B shows a subset of the mask images for the more isotropic matrix. The absolute metrics for filament length, end points and branch points scaled with the size of the input image in pixels–this is also clearly visible in the masks that are generated, with reduced matrix network complexity apparent as the number of pixels is reduced. Normalisation of the filament length to the image size and the end points and branch points to the total filament length eliminated the relationship between image size (in pixels) and length, end points or branch points (Table S1 and Fig S5). When using two-dimensional images, it is possible that two fibres in different focal planes might be erroneously interpreted as branching. The use of physical sectioning when carrying out histopathological stains and optical sectioning when using confocal methods will minimise this. Nonetheless, we compared the number of branch points identified in a single confocal section with a maximum intensity projection of seven confocal sections (equivalent to increasing the sample thickness by a factor of seven). For the isotropic matrix, the normalised branch point value varied by less than 4%. The divergence in values for the aligned matrix was 16% (values are highlighted in green in Table S1). This confirms that using input images with a consistent thickness of section is important, but also indicates that the level of erroneous branch point identification is modest even if the thickness of section varies sevenfold. It should be noted that the normalised end point metric is the inverse of the HGU metric. Therefore, we use normalised versions of these metrics in subsequent analyses and would recommend that other users do the same.

Having determined which metrics required normalisation and which did not, we then asked whether TWOMBLI could reliably quantify differences between the isotropic and anisotropic CDMs. To obtain a broad overview of the data, we generated a PCA plot using the normalised and scale-free metrics, with the exception of curvature that exhibited high variability. Fig 4A shows that the quantification of the isotropic CDM and anisotropic CDM largely fall into two distinct clusters; however, there are some outliers and the separation is not perfect. Further inspection revealed that the most marked outliers were under-exposed images or those with added Gaussian blur or noise (Table S1). If the analysis was restricted to images captured using the same 20× objective, with appropriate exposure, the same pixel size, and same line width value, then TWOMBLI generated two well-segregated clusters in PCA analysis corresponding to the isotropic and anisotropic matrices (Fig 4B). Analysis of individual variables revealed that many were remarkably robust to the image acquisition settings, with alignment, fractal dimension, and lacunarity being particularly insensitive to variation in microscope settings and even the use of a 10× or 20× objective (Fig 4C and Table S1). Curvature was the one parameter that exhibited a high degree of variation. Smaller curvature windows are able to capture waviness of fibres at a finer scale, whereas larger windows describe curvature of more macroscopic fibre structures. To this end, TWOMBLI allows users to specify a minimum and maximum curvature window. Curvature at

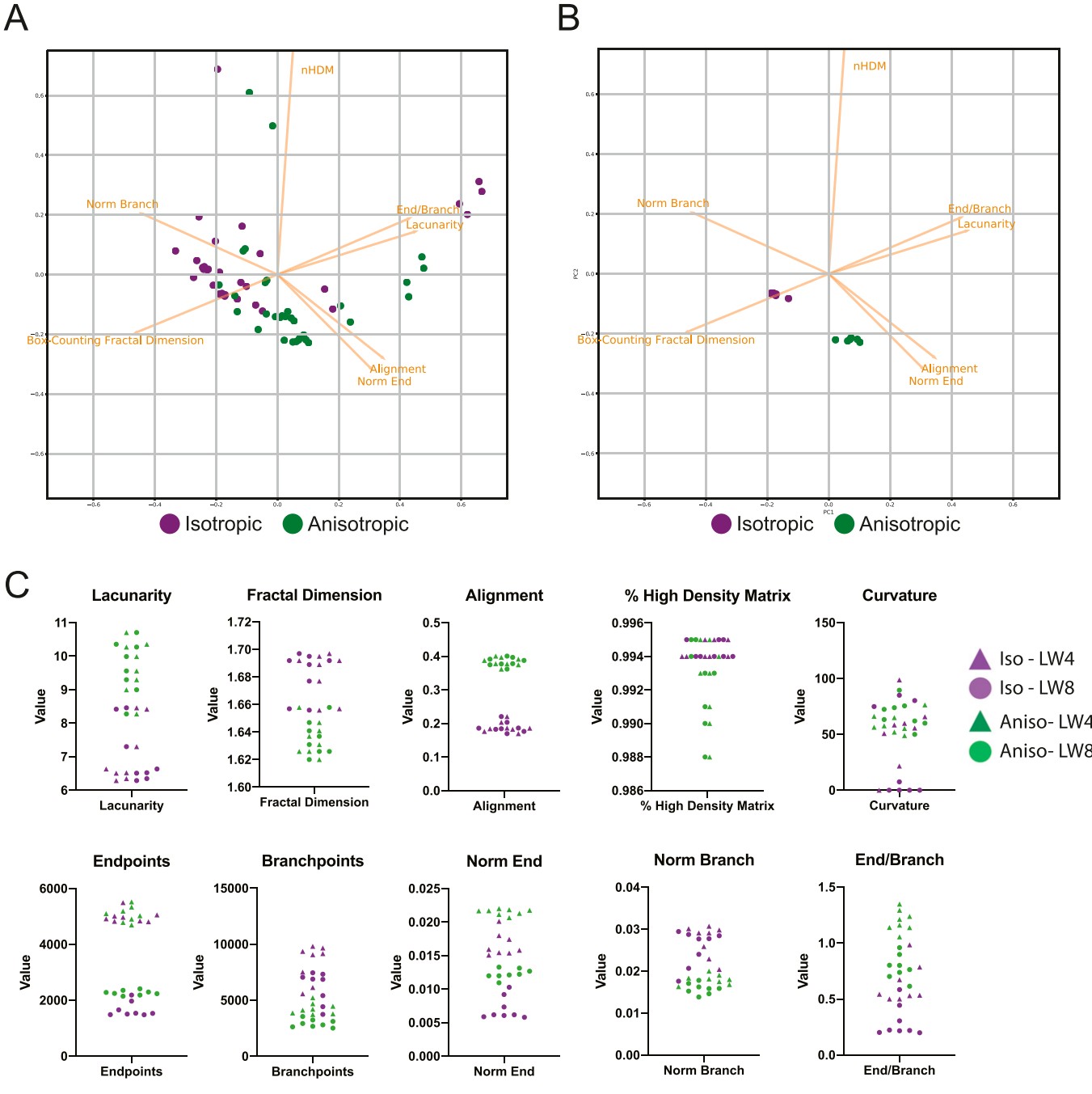

**Figure 4. Robustness of matrix metrics.**
**(A)** Image shows a PCA plot of both experimental and artificial variations in images of the same region of isotropic cell-derived matrix (CDM) (purple dots) and anisotropic CDM (green dots). **(B)** Image shows a PCA plot of metrics derived from images captured with a 20× objective of uniform pixel size, exposure, and line width of the same region of isotropic CDM (purple dots) and anisotropic CDM (green dots). **(C)** Plots show the values of the indicated metrics with isotropic CDM indicated in purple and anisotropic CDM labelled in green. As in (B) only images with the same pixel size and appropriate exposure are included, but images captured with a 10× objective and more divergent focal planes are also included. Furthermore, two different line width settings are shown: triangles denote use of line width 4 and circles use of line width 8. Corresponding images can be found in Fig S4 and a full corresponding list of metrics is given in Table S1

different window values within this range is then computed. Closer inspection of the details of pixel size and line width selected revealed that the normalised end point metric was sensitive to the choice of line width, with more nuanced variation in other metrics depending on the pixel size and line width choice. For optimal robustness, we would recommend that users select a single line width/ range of line widths for their analysis and do not vary this. Overall, these data indicate that the metrics generated TWOMBLI are robust to the precise choice of microscope objective, pixel size, and line width. The biggest factor leading to variation in output metrics was incorrect image exposure. The precise choice of focal plane for imaging and its thickness had relatively little effect. Minimal variation

in metrics was achieved with a consistent image resolution and line width choice.

**Output metrics are able to distinguish between different matrix patterns**

Having determined that TWOMBLI is able to generate reliable metrics, so long as simple principles of consistency in image acquisition and analysis parameters were followed, we tested whether TWOMBLI could provide a quantitative framework for discriminating between different ECM architectures. To this end, images of CDMs generated by seven different isolates of fibroblasts were analysed, with multiple images for each example. Fig 5A shows that TWOMBLI could effectively identify matrix fibres. Furthermore, the alignment metric accurately separated the CDMs which had previously been classified as either isotropic or anisotropic using a more complex MATLAB tool (12). TWOMBLI also revealed notable differences in addition to alignment, the larger gaps in the MAF2 matrix were reflected in the lower fractal dimension and higher lacunarity (Fig 5B and Table S2). As might be expected, isotropic matrix patterns generally had lower levels of normalised branch points. PCA analysis of the different CDMs is presented in Fig S6. This shows clear separation of the aligned and non-aligned ECM but also highlights the distinct nature of MAF2-derived matrices, which have greater lacunarity.

Quantification of matrix from the breast cancer biopsies presented in Fig 2 shows how some metrics such as HDM are intuitively different, whereas other metrics such as curvature and branch points are more nuanced (Fig 5C). We deliberately reuse the images from Fig 2 to demonstrate that qualitative pattern features readily discernible can be quantified by TWOMBLI, and importantly, that TWOMBLI could be particularly helpful in identifying patterning properties that are difficult to ascertain by eye. The application of TWOMBLI to prostate biopsies is demonstrated in Fig S7A, with quantitative differences in HDM, fractal dimension, and lacunarity collectively distinguishing tumour regions from both normal glandular and normal stromal regions of prostate tissue (Fig S7B and Table S3). Higher lacunarity and lower fractal dimension are observed in the glandular areas of normal prostate, which reflects the absence of large ECM fibres within the epithelial compartments and highlights the importance of metrics besides alignment and orientation. Taken together with the robustness results, these analyses suggest that TWOMBLI is able to distinguish between clearly different matrix patterns.

Finally, we sought to test whether our pipeline might have utility beyond the analysis of ECM fibres. Dynamic filamentous polymers, including F-actin and microtubules, within cells collectively make up the intracellular cytoskeleton. Fig S8 shows images of the F-actin network within the NF2 and CAF1 fibroblasts used to generate the CDMs that we analysed in Fig 5. We have previously reported increased F-actin stress fibres in CAF1 and, consistent with this, the TWOMBLI pipeline correctly identifies an increased length of filamentous structures. We also sought to investigate whether TWOMBLI could quantify super-cellular patterns of organisation, such as vascular networks. Fig S8A shows that TWOMBLI can indeed identify fluorescently stained endothelial cell networks in the mouse embryo. Furthermore, analysis of different regions shows

that the network density is reflected in differing fractal dimension, branch point, and lacunarity scores, which re-iterates the utility of these metrics. This demonstrates that our analysis tool could analyse diverse imaging data spanning sub-cellular filamentous networks through to the ECM in pathological samples.

# Discussion

Much work has focused on identifying the distribution and patterns of cells in tissues (22, 23, 24). However, many studies do not fully document the organisation of the ECM, which plays a crucial part in tissue architecture. We have developed TWOMBLI to quantify a wide range of matrix features, enabling further understanding of the relevance of ECM organisation in a wide range of contexts, from tissue damage (25, 26, 27) to ageing (28), development (29) to fibrotic disease and cancer. This tool is designed to generate a wide range of metrics that capture diverse aspects of matrix pattern in the same pipeline, thereby saving researchers from having to run multiple analytical tools in parallel on the same images. Nonetheless, the derivation of certain more complex metrics, for example, the relative angle of collagen fibres to tumour nests, still requires more bespoke tools (13). FIJI was chosen as the platform because it is freely available, well known and widely used by the biological and medical sciences community (4, 30).

Our analysis indicates that the metrics generated are relatively robust to both the precise region of matrix selected and to the exact microscope settings (Fig 4). Sub-optimal exposure of images was the most detrimental factor in terms of the robustness of the output metrics generated and we would advise researchers to ensure appropriate exposure settings. The objective used and pixel size should be kept constant for optimal results. Slight variations in focal plane or the thickness of the optical section had only minor effects on the output metrics (Fig 4B). Attention should be paid to the ridge detection width because this parameter dictates whether the algorithm identifies fine or thick matrix fibres. Analysis of the same images using two different line widths revealed that the identification of end points and branch points was sensitive to the choice of line width. In this case, TWOMBLI is able to identify fibres over a range of line widths. However, many parameters were relatively insensitive to this setting, which should enable comparison of metrics between different studies conducted using slightly different analysis settings. The only parameter that we would advise extra caution to be taken over is curvature, which showed a high degree of variation. For this reason, specifying different minimum and maximum curvature windows will enable the user to capture both finer and coarser descriptions of curvature. In some cases, it may be advantageous to run the same images with two different ridge detection widths so that both fine and coarse features can be captured.

The tool that we describe here is designed to be simple to use and generate a wide range of metrics. It is not designed to be highly specialised in its handling of particular matrix features. Excellent specialist tools exist for researchers who may wish to dig deeper into particular features of the ECM; in particular, the Eliceiri group has developed a suite of tools to interrogate matrix alignment using MATLAB (13), including analysis of matrix fibre orientation relative to tumour cells. Our group has also generated more bespoke tools for measuring matrix alignment over a range of length scales and this

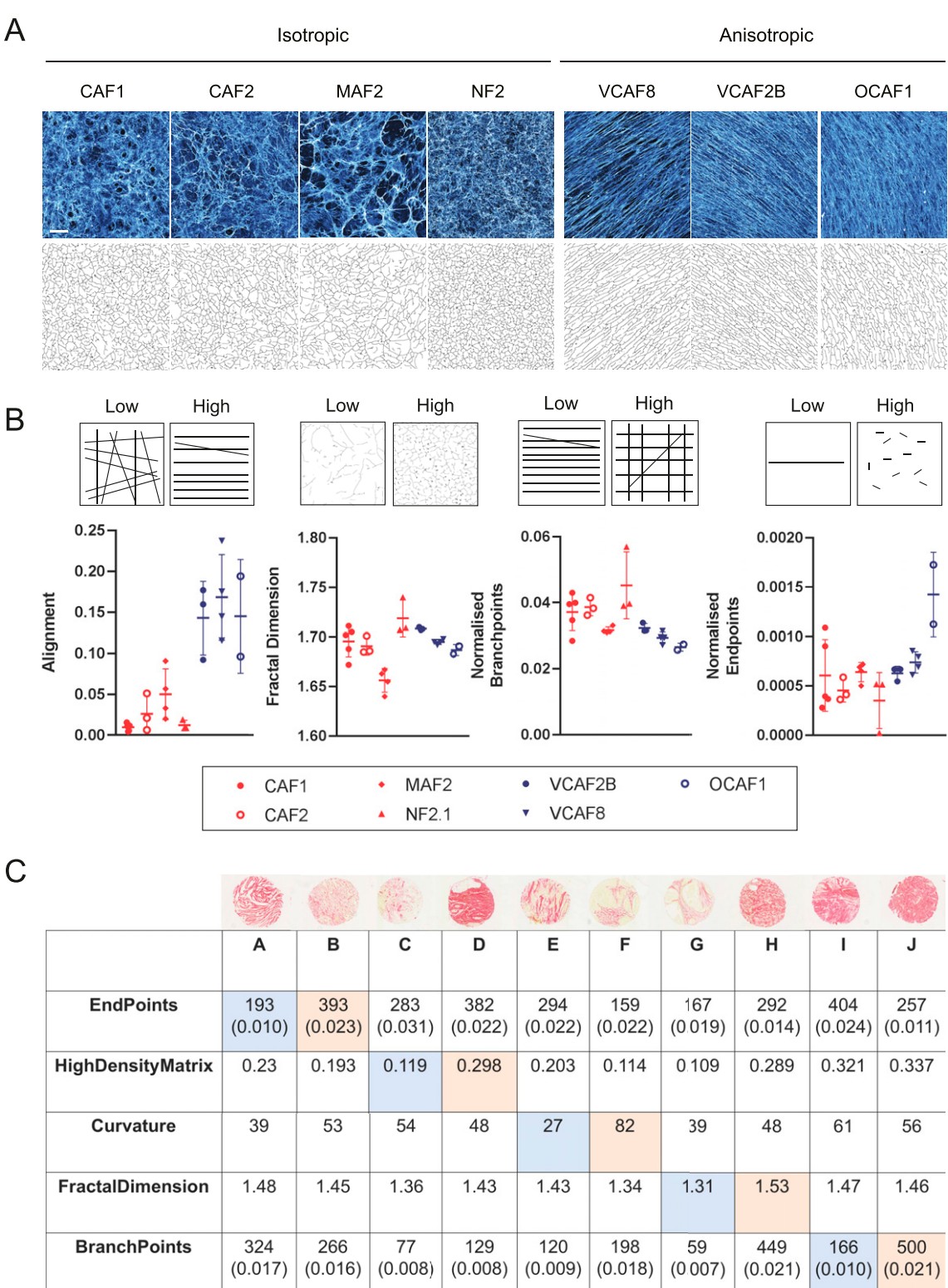

Figure 5.  Using TWOMBLI for quantification of ECM patterns.
**(A)** CDM imaging of fibronectin produced by seven different fibroblast lines in vitro (4). Four patterns are isotropic (left) and three patterns are anisotropic (right). Images are 500 × 500 $\mu$m, scale bar is 100 $\mu$m. Corresponding masks derived in TWOMBLI are displayed underneath. **(A, B)** Quantification of matrix patterns from (A) across a number of metrics. Normalisation is performed by dividing the raw value by the total length of fibres in the masks. Full list of metrics given in Table S2. **(C)** Output metrics for the corresponding tumour biopsies shown in Fig 2. Pairs (A, B), (C, D) etc. show contrasting biopsies with low and high values of each metric as indicated in the table in blue/orange, respectively. For end points and branch points, normalised values are given in brackets below the raw value. The normalised value is computed by dividing the raw value by the total length of the fibres in the mask.

may be of use to researchers wishing to compare short-range and long-range matrix alignment. The alignment metric in TWOMBLI is not capable of this level of sophistication and simply generates a global alignment score for the whole field of view. MATLAB tools for the analysis of matrix gaps have also been generated (30). Haralick features have also been used to measure ECM organisation (7); however, we did not include these because textural analysis is not well suited to our filament-based approach. Nonetheless, Haralick analysis may be a useful addition to future iterations of matrix analysis platforms, especially if performed on images before filament tracing.

The tool that we describe has been developed with the purpose of quantifying ECM fibres, and it can also be applied to other classes of biological images, such as networks of F-actin or microtubules or even vascular networks. Its broad applicability is evidenced by its ability to analyse the F-actin cytoskeleton of the fibroblasts and vascular organisation of the mouse embryo (Fig S8). Optimal utilisation for diverse classes of images is likely to require some additional adaptation that account for differences in the input data; for example, sub-classification of arterial and venous vessels in vascular networks. Furthermore, the ability of this tool to segregate normal prostate images from tumour regions suggests that it could be exploited in the analysis of clinical samples. It is our sincere hope that this versatile tool will be of use to cell biologists, tissue biologists, and pathologists, enabling these communities to study the consequences of different matrix architectures in disease.

# Materials and Methods

### Computational analysis

All computational methods are described in detail in the TWOMBLI documentation which can be found at https://github.com/wershofe/TWOMBLI.

### Picrosirius red staining

Samples were stained using ABCAM ab150681 Picrosirius Red Kit. Briefly, slides were deparaffinised and hydrated, before applying Picrosirius red solution for 60 min, rinsing twice in acetic acid, then alcohol dehydration, and finally mounting. Slides were then scanned at 10× using Zeiss Axio Scan.Z1.

### CDM assay

The CDM assay and the fibroblasts used are described in Park et al (4). Briefly, glass-bottom dishes (P35-1.5-14-C; MatTek) were pre-prepared with 0.2% gelatin solution (1 h, 37°C), then 1% glutaraldehyde for 30 min at RT. After PBS buffer solution wash, the plate was incubated with 1 M ethanolamine for 30 min (RT). After two washes with PBS, we seeded $7 \times 10^4$ cells in media with 100 $\mu g$ ml$^{-1}$ ascorbic acid ((+)-sodium L-ascorbate, A4034; Sigma-Aldrich). Cells were kept for 6 d and media changed every 2 d. We used extraction buffer and washed several times with PBS before immunofluorescence for ECM using anti-fibronectin-FITC (1:50 dilution, ab72686;

Abcam). Samples were imaged using a Zeiss LSM 780 microscope using either a 20× 0.75 NA objective or 10× 0.45 NA objective.

### Collagen imaging

Second-harmonic generation imaging for collagen was performed as described in Park et al using a Zeiss LSM 780 microscope with Mai Tai multi-photon laser. Fresh post-mortem tissue from PDGFRA:: H2B-eGFP mice was used.

# Supplementary Information

# Acknowledgements

We thank the members of the Sahai group, Panoraia Kotantaki, Florian Laflorêts for advice and comments. We thank Prof. David Dearnley, Prof. Emma Hall, Dr Michael Toss, and Dr Emad Rakha for assistance with clinical samples. E Wershof, D Park, DJ Barry, A Wilkins, RP Jenkins, KI Anderson, PA Bates, and E Sahai were funded by the Francis Crick Institute, which receives its core funding from Cancer Research UK (FC001003, FC010144), the UK Medical Research Council (FC001003, FC010144) and the Wellcome Trust (FC001003, FC010144). A Rullan was supported by the Spanish Society for Medical Oncology (Beca Fundación SEOM). E Sahai and D Park also received funding from Breast Cancer Now (2013NovPR182). A Wilkins is also supported by a Crick i2i grant.

### Author Contributions

E Wershof: conceptualization, formal analysis, methodology, and writing—original draft, review, and editing.
D Park: formal analysis and methodology.
DJ Barry: methodology.
RP Jenkins: methodology and writing—review and editing.
A Rullan: methodology and writing—review and editing.
A Wilkins: formal analysis, methodology, and writing—review and editing.
K Schlegelmilch: data curation.
I Roxanis: data curation.
KI Anderson: project administration.
PA Bates: conceptualization, supervision, and writing—review and editing.
E Sahai: conceptualization, formal analysis, supervision, project administration, and writing—original draft, review, and editing.

### Conflict of Interest Statement

The authors declare that they have no conflict of interest.

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
