## [Reviewer comments · Life Science Alliance]

Life Science Alliance

A FIJI Macro for quantifying pattern in extracellular matrix

Esther Wershof, Danielle Park, David Barry, Robert Jenkins, Antonio Rullan, Anna Wilkins, Ioannis Roxanis, Kurt I. Anderson, Paul Bates, Erik Sahai, and Karin Schlegelmilch

DOI: <https://doi.org/10.26508/lsa.202000880>

Corresponding author(s): Erik Sahai, Cancer Research UK - London

Review Timeline:	Submission Date:	2020-08-17
	Editorial Decision:	2020-08-18
	Revision Received:	2020-11-24
	Editorial Decision:	2021-01-05
	Revision Received:	2021-01-07
	Accepted:	2021-01-08

Scientific Editor: Shachi Bhatt

Transaction Report:

Please note that the manuscript was previously reviewed at another journal and the reports were taken into account in the decision-making process at Life Science Alliance.

August 18, 2020

Re: Life Science Alliance manuscript #LSA-2020-00880-T

Dr. Erik Sahai
Cancer Research UK - London
Cancer Research UK
Centre for Cell & Molecular Biology
Institute of Cancer Research
London, 237 Fulham Road UK-London SW3 6JB
United Kingdom

Dear Dr. Sahai,

Thank you for submitting your manuscript entitled "A FIJI Macro for quantifying pattern in extracellular matrix" to Life Science Alliance (LSA). The manuscript was assessed by LSA editors and deemed appropriate for publication at LSA with minor revisions.

For a brief overview: This manuscript was peer-reviewed at another journal and the editors of that journal transferred the referee reports with authors' permission to LSA. The main criticisms raised by the referees that precluded publication in the previous journal pertained to lack of technical advance and absence of proof-of-principle (pop) analysis. Given the ability of the new tool defined in this study to make ECM quantification more accessible to researchers, and the value of ECM quantification in various pathologies and aging, the manuscript was considered to provide sufficient advance for LSA.

Proof-of-principle analysis requested by the reviewers will not be required for publication at LSA. We encourage you to re-submit your revised manuscript to LSA with the following technical edits:

- + Please provide a point-by-point response to all the concerns raised by the reviewers at the previous journal
- + The abstract needs to be re-written to include information about what parameters this tool quantifies, the advantage(s) offered by this tool and clarify that the analysis is specific to fibrous matrix (R1 pt 3)
- + Please provide explanations, and data to respond to the concerns raised by the referees about collagen fiber diameter or thickness, waviness of the fiber, and differentiation between branchpoints vs. two fibers in separate z-planes (R1 pt 4,5,8; R2 pt 1, 2 and some minor points)
- + Please discuss differences in processing colorimetric stainings vs fluorescence images (R1 pt 7)
- + Please provide higher quality / high res zoomed in images in Fig 2 & 3 (R1 pt 6)
- + Please revise the text to make the manuscript more understandable to generalist readers and potential users (editorial request), and emphasize how the selected metrics used in this tool are an improvement over what is currently available or how the tool is more accessible than what is currently available (as pointed by R2)
- + Please also comment on or discuss the low sensitivity of the parameters in this tool, which requires user input, and the caveats of doing so (pointed out by Rev 2)
- + All the other minor concerns, edits requested by the reviewers should be addressed in the revised manuscript too

Thank you for this interesting contribution to Life Science Alliance. We are looking forward to receiving your revised manuscript.

Sincerely,

Shachi Bhatt
Executive Editor
Life Science Alliance

B. MANUSCRIPT ORGANIZATION AND FORMATTING:

Dear Dr Lorenz and the LSA editorial team,

We are pleased to resubmit our manuscript "A FIJI Macro for quantifying pattern in extracellular matrix" to Life Science Alliance. Following the advice of the reviewers, we have made several changes and improvements to our manuscript. These include:

- Modifications to allow automatic analysis of multiple line widths and curvatures
- Methods for the estimation of fibre length and width
- Improved description of the metrics and their usage
- Explaining how the tool is specific for branchpoints
- Explaining how to analyse coloured histochemical images
- Demonstration of the utility of the tool for vascular analysis
- Providing larger and higher magnification images
- Text editing for improved clarity

A detailed point-by-point response is included below and all changes in the documentation are in green. We believe that these changes have improved the TWOMBLI tool and that it will be an asset for researchers and clinicians interested in the extracellular matrix and, more broadly, anyone studying filamentous structures in biology.

Reviewer #1 (Comments to the Authors (Required)):

In this Tool manuscript, Wershof and colleagues describe a Fiji Macro for quantifying patterns in extracellular matrix. The Macro takes input images and quantifies various characteristics including curvature, fractal dimension, number of branch points, total length, number of end points, lacunarity, alignment and gap analysis. There is certainly significant interest in automated analysis platforms of extracellular matrix architecture, and improved analysis tools would potentially be very impactful. However, there are a number of major questions concerning the utility, novelty, and specific features of the introduced tool that need to be addressed before suitability for publication in the Journal of Cell Biology can be assessed.

We are pleased that the reviewer comments that "there is certainly significant interest in automated analysis platforms of extracellular matrix architecture, and improved analysis tools would potentially be very impactful".

1. The biggest missing item, to me, is a proof-of-principle analysis that demonstrates the utility of this tool for the cell biology community. The analysis of isotropic vs. anisotropic CDM is shown in Figure 3 and 4, but this is not very interesting in itself, and its not an example that takes advantage of the full capabilities of this tool. Devising an algorithm to show differences between isotropic and anisotropic matrices is simple, and doesn't

require most of the different metrics. A more compelling proof of principle analysis should take advantage of all the different analysis threads and show something potentially interesting. The analysis of prostate cancer in supplementary figure 4 starts to get at this, but the analysis is minimal. One possibility is that the authors could do a much more robust analysis of prostate cancer: for example, histograms of all of the different metrics for a sufficiently large set of images of tumor vs. normal, etc with statistical analysis. Or, perhaps there is another such analysis that could be done based on available pathology images. Whatever it is, some stronger proof of principle analyses are needed to demonstrate the utility of this tool to the reader.

We agree with the idea that the reviewer is proposing and have recently received funding to explore in some depth the associations between ECM metrics and clinical features in both breast and prostate cancer. However, this is a substantial endeavour and will be appropriate for a clinical journal. The intention of this technical report is to establish the methodology and make it available to the research community. Following advice from the LSA editorial team, we are not including this type of analysis in the re-submitted manuscript.

2. Novelty: it is difficult to determine what the specific advance of this current tool is. From my read of it, it seems like the algorithms for each specific output are already established, and the advance is just in the combination of all tools in one pipeline - is this correct? As the criteria for a JCB tool is that they "should describe a technological advance of broad/general interest that permits the interrogation of cell biological problems in ways previously impossible ...". In the absence of #1 or #2, it wouldn't seem like this manuscript meets this criteria.

The intention of our TWOMBLI tool is to promote the widespread and consistent use of metrics for fibrillar patterns in biology and medicine. This is an 'unmet need' with the majority of papers in this area either concentrating on one or two metrics to the exclusion of others, or not providing quantification at all. The reviewer is correct that we have not invented a new metric, but this was not our goal. The novelty of the tool lies in the diversity of metrics that it generates and therefore the ability to describe complex patterns more precisely in multidimensional space. Furthermore, as this work is now being considered for LSA the issue of suitability for JCB is no longer relevant.

3. The abstract is very vague and leaves the reader with little idea of what parameters this tool quantifies and what the advance of the tool is - these should be specified. Further, this analysis is specific to fibrous matrix - that should also be specified in the abstract.

We have now added two sentences to the abstract - lines 22 - 29: one lists key metrics and the other indicates the types of images that can be analysed.

4. Collagen fiber diameter is an important feature of ECMs, which does not appear to be analyzed by this pipeline. Why?

We agree that this is an important feature of the ECM. The simplest solution is to divide the high-density matrix value (effectively the area) by the total fibre length. This provides an estimate of the average fibre width. The efficacy of this method is shown below.

In addition, we have now implemented the option to analyse multiple fibre diameters simultaneously. Specifically, the user can choose a minimum and maximum line width value. If they choose these values to be the same (default setting) it is equivalent to running a single line width for ridge detection as per the previous version of TWOMBLI. If the maximum line width value is greater than the minimum, the ridge detection will be run for line width values between the minimum and maximum with step size one. This will generate a mask picking up fibres of multiple thicknesses, with the caveat that multiple line widths will slow down computation. This is explained in the documentation and main text on lines 110-112 and 228-236. The resultant data can be used to analyse how matrix metrics varying as a function of line width.

5. How does the pipeline differentiated between branchpoints vs. two fibers that are not connected and in different z-planes?

The reviewer raises a valid point. The typical input images envisaged for analysis are either physical sections or optical sections. These have only a very limited z dimension relative to their x and y dimensions - 4-5 microns for histological sections and 1-2 microns for confocal sections. This means that the issue of two fibres being in different z planes but being interpreted as a branchpoint will be relatively rare. Nonetheless, the reviewer's comment is well made. To examine this issue we have directly compared the number of normalised branchpoints detected if a single confocal section acquired with a narrow pinhole, a single confocal section acquired with a wide pinhole, and z-stack of confocal sections are compared. This is now described in detail in lines 257-267 and the metrics are highlighted in green in Table S1.

6. The zoomed out images in figures 2 and 3 make it hard to see the individual fibers and see if the processed image faithfully captures the fibers. Zoom in images should be shown for both.

We agree with this comment. The new Supplementary Figure 1 shows both Picrosirius Red and Masson's Trichrome images and their corresponding masks at sufficient magnification to be able to relate the input image to the mask. The revised Figure 3 now includes zoomed in images showing an overlay of the input image and the mask. This clearly shows the relationship between the input data and the mask that is derived.

7. The tool analyzes both colorimetric stains as well as fluorescence images. The authors should discuss any differences in processing for these two different inputs/ any differences in the quality of the analysis.

We agree that this is an important issue and we have addressed it by adding a new Supplementary Figure 1. In summary, to extract matrix patterns from histochemical stains, such as Picrosirius Red (PSR) or Masson's Trichrome (MTR), we recommend

using the Color Deconvolution tool in ImageJ (Figure below). To negate variability between staining batches and different microscopes, we advise using the ROI tool to deconvolute the matrix stain from conventional RGB images. The top panels show RGB images of serial sections of mouse prostate stained with PSR (left) and MTR (right). For the PSR image, ROIs were selected corresponding to the red matrix stain, brown nuclear color, and gray background (indicated with yellow boxes). For the MTR images, ROIs were selected corresponding to the blue matrix stain, purple nuclear color, and gray background. Middle panels show the deconvoluted matrix stain for PSR (left) and MTR (right). Lower panels show the masks generated from the deconvoluted matrix images. For large batches of images, we recommend deconvoluting a small number manually using the ROI method and then generating an average weighting matrix using the outputs of the channel weighting for each test image produced by ImageJ. This matrix can then be entered into the Color Deconvolution tool and applied to an entire batch of images.

8. Collagen fibers can have curvature over different lengthscales - for example, collagen fibers can often be "crimped" or ruffled and have both a short wavelength and long

wavelength curvature. How does the algorithm handle such scenarios? Over what length-scales is processing of curvature tuned for?

The reviewer raises an interesting point. To test the effects of different curvature lengths we manufactured 'fibres' by creating sine curves with various amplitudes and frequencies (see figure below) which were run in TWOMBLI with a range of curvature windows. The table at the bottom of the figure shows that curve A, with low amplitude and frequency has low curvature value over all curvature windows, B & D which have high frequency but low amplitude, representing curves with high tortuosity have high curvature for low curvature windows, whereas E & F which have high amplitude have higher curvature for larger windows, capturing curvature at a coarser scale. This suggests it would be pertinent to have a range of curvature windows to pick up both coarser and finer curvatures. We have since modified TWOMBLI to allow the user to select a minimum and maximum curvature window. TWOMBLI will then analyse curvature in the range of these two values with increments of 10 pixels. The smaller values are intended to capture curvature at a finer scale whilst the larger values pick out more global curvature in the image. In the output results file, the smaller and larger curvature metrics are outputted in separate columns.

Label	Curvature_10.0	Curvature_20.0	Curvature_30.0	Curvature_40.0	Curvature_50.0	Curvature_60.0	Curvature_70.0	Curvature_80.0
A	6.785	12.726	17.578	20.45	21.308	20.949	20.217	19.781
B	29.772	47.276	45.869	29.812	9.032	0.496	4.128	12.514
C	12.977	25.852	36.433	42.903	46.156	46.773	45.873	45.656
D	25.773	50.649	70.007	79.085	79.624	74.664	62.002	42.038
E	12.36	23.867	36.687	48.002	56.363	62.526	66.715	68.531
F	15.539	30.589	46.214	60.745	73.562	84.638	93.361	100.159

This is now explained in the documentation and main text on lines 240-243 and 320.

Reviewer #2 (Comments to the Authors (Required)):

Wershof et. al., entitled "A FIJI Macro for quantifying pattern in extracellular matrix" aims to provide a tool (i.e., publicly accessible ImageJ plugin) for quantitative assessments of extracellular matrix (ECM) images, aimed to be user friendly. The tool (i.e., WOMBLI) simultaneously reports on a plethora of architectural parameters. The provided rationale to justify a need for this tool is that available tools are either too complicated for the non-

computational biologists or exclusively centered on a limited number of parameters. The WOMBLI plugin is a bit involved, but accessible and capable of measuring topographical nuances that may not be obvious to the human eye. Nonetheless, the authors failed to emphasize how their selected metrics improve on the ones available (i.e., better convey pathophysiological instances that could justify the metrics selections). For example, it is clear and has been shown that tumor associated collagen signatures (TACs) and ECM fiber alignment are metrics associated with patient outcomes as these serve as physical paths that increase metastatic probabilities. The question here is: how the new compiled metrics are to advance on available information (i.e., what added information is the new HGU metric providing)? Or if the tool provides the same information, how is it more accessible? This type of issue constitutes an addressable pitfall that merely needs added attention by the authors. Another point for improvement is the apparent low sensitivity of the parameters, needing user input, which may not account for the true range of ECM fiber heterogeneity. If these points are addressed together with the minor listed points provided below, intended to assist the reader to improve on the didactic instruction and depiction of the tool, the potential impact of the manuscript could be greatly enhanced.

Major points:

1. A potential technical caveat of how TWOMBLI works is the fiber thickness input. It may be better if the user were to provide a range of thickness as these greatly vary from sample to sample and even within the same sample. As a user, one should not need to compromise between resolution of fiber detection and being able to also account for thick fibers. The software right now fails to distinguish between fiber bundles and a true thick fiber, while thin fibers are undetectable (and vice versa).

The reviewer is correct that the thickness of fibres is important - reviewer #1 raised a similar point. We have now implemented the option to analyse multiple fibre diameters simultaneously. Specifically, the user can choose a minimum and maximum line width value. If they choose these values to be the same (default setting) it is equivalent to running a single line width for ridge detection as per the previous version of TWOMBLI. If the maximum line width value is greater than the minimum, the ridge detection will be run for line width values between the minimum and maximum with step size of five. This will generate a mask picking up fibres of multiple thicknesses, with the caveat that multiple line widths will slow down computation. This is explained in the documentation and main text on lines 228-236. The resultant data can be used to analyse how matrix metrics varying as a function of line width.

Additionally, if the user wishes to generate a mask based on the amalgamation of masks generated using different line widths this is now possible, although it is computationally expensive.

2. Similarly to the point above, input values for curvature could also include a range as opposed to a single value (again due to the heterogeneity seen in the samples, including the ones presented in the training samples).

The reviewer is correct that multiple curvatures could be informative - reviewer #1 raised a similar point which we have addressed above.

3. Note that similar predicaments could be noticed with regards with other parameters such as "gaps," etc.

The gap analysis algorithm iteratively fills the largest possible circles in empty space, incrementally fitting smaller and smaller circles down to the user-specified minimum circle diameter. Therefore, there is no need to specify multiple gap values. The user should be able to determine a reasonable minimum diameter below which the gaps are negligible. This is similar for the HDM metric. The user is required to use their judgement, but it is hoped that thorough documentation and the tutorial video will enable sensible parameter choice in these cases.

Minor points:

- The introduction could benefit from adding a stronger explanation of ECM function and known link between its topography and putative roles; this will help to improve on the rationale needed to assertively justify the requirement for developing such an intricate digital tool.

We have now added some sentences in the introduction highlighting work suggesting that the linearity and alignment of collagen fibres is associated with an ability to resist tensile stress and more aggressive cancer phenotypes - lines 45 - 48.

- For the biologist reader that is not a biophysicist, the introduction can also be supplemented with a simple and brief explanation for the gray level co-occurrence matrices as well as fractal dimension (in the second paragraph of the introduction).

We have added this information to the introduction - lines 46 - 48.

- Please briefly explain to the reader what the existing plugins that the TWOMBLI builds on are about.

Ridge detection is an algorithm for detecting the filamentous structure of an image in an unbiased manner. Anamorf is a plugin for quantifying such structures, originally developed for describing the growth of mycelium. TWOMBLI uses many of these metrics and adds others more pertinent to the study of matrix fibres. For example, OrientationJ provides the global alignment of the structure, which is not provided by Anamorf but is important to know in the context of fibre organization. We have added in such a description in lines 84-88.

- It is obvious to this reader that the more images one intends to analyze the longer computational time it will take to process (and many users may not have fast computers). It is also clear that while the visualization of the masks and other outputs are very informative, the user will very rarely need ALL and probably (based on representative numerical outcomes) will be happy to select a few images and generate

the mask and other visual outcomes solely from selected representative ones (perhaps in a new run of the plugin). Hence, if numerical outputs are to be often the main user of the goal, the tool could have an option to opt out of image outputs.

The reviewer is quite correct that computational time can become an issue when analysing large numbers of images. The main functions of TWOMBLI work in batch mode, taking as input the folder containing the image masks. In response to the reviewer's suggestion, we disabled the generation of output images and compared the time taken for analysis. This revealed that there was negligible time saving. The reason for this is because the masks still need to be generated for the analysis to proceed and therefore the only saving is the time taken to save the images to a local hard drive, which is considerably less than a second for a typical image of 1-10 Mb. We also verified with several examples that saving the images is computationally much faster than the processing within TWOMBLI ie removing any image outputs will minimally save on time.

- Similarly to the point above, the initial steps for setting up the parameters (including visual outputs) could represent all what a user wants to do. In this case numerical outputs should be an option for a one or two image analysis. This user understands that the initial steps can be spared if the parameters are identical and have been pre-set but this portions/aspect was not immediately obvious.

Regrettably, we are a little unsure exactly what is requested in this comment. Our understanding is that the reviewer is asking about how the optimal parameters are determined and whether they can be applied in batches. We recommend determining the optimal parameters on a small subset of images. These should cover the diversity of images in the full set. Parameters should be optimised so that the user is happy with the mask images generated. It is true that a small computational time saving could be achieved if all the output metrics are not generated during this part of the process.

Once satisfied with the parameter selection, the user should then save the parameter file and use this for the whole batch of images.

- What is a spatial derivate (please provide an example)?

We believe the reviewer is asking about spatial derivatives, we have now added a sentence in line 93 explaining that spatial derivatives measure the rate of change in intensity value between nearby pixels.

- Please provide references for the alternative approaches on this statement: "making the algorithm far more computationally efficient than alternative approaches"

This comment was based on our empirical experience using both CT-fire and our pipeline, hence there is no existing reference to cite. To clarify for the reviewer, we find that, compared to CT-fire, our ridge detection runs at least 10 times faster on the PSR images. However, we realise that without a reference or further evidence the statement

in the text is not entirely justifiable. Therefore, we have removed the part of the sentence comparing our algorithm to others - line 85.

- Perhaps the word "metrics" (or similar), is missing from this sentence: "We selected that covered different aspects of ECM pattern"

Thank you for pointing this typographical mistake, we have now rectified it - line 107.

- Just like assertively done for hyphal growth unit, the authors may want to define "fractal dimension" adding something like "a metric depicting the level of complexity of the shape being analyzed" or similar.

This is a good suggestion and similar to one made by reviewer#1. We have now added a definition - lines 47 & 48.

- Average fiber length, in addition to the provided sum, could constitute an added important output (i.e., as it could inform on the potential forces ECMs could withstand).

This is a good point. We now state in the text that the average fibre length can be calculated from the total fibre length divided by $0.5 \times$ (the number of endpoints + branchpoints) - lines 131 - 133.

- Important published/reported parameters that correlate with patient outcomes are the waviness vs. straightness of the fibers, which is different from the described "curvature" as the mean angle (or intra-fiber orientation) will be kept mostly constant. Could the authors comment on this and explain the user whether curvature output could (or could not) serve as proxy?

The reviewer is correct that waviness vs straightness is an important distinction. As described in the point and figure above, we have now implemented multiple curvature windows in TWOMBLI, where the smaller curvature windows are able to capture the 'waviness' of the fibres and larger curvature windows report on larger scale curvature.

- Similarly, it is unclear which parameters provided by the plugin are to be compared to alignment and orientation that are the ones most commonly used today. Also, which metrics are to inform on TACs1 vs TACs2/3 (it is clear that they don't encompass means to distinguish between TACs2 and TACs3).

The metrics generated by TWOMBLI are not intended to duplicate the information contained in Tumour-Associated Collagen Signatures (TACS). Nonetheless, the reviewer raises an interesting question. Broadly, TACS1 is informed by HDM (high-density matrix) and TACS 2 is informed by the alignment metric. TACS3 is not encapsulated by TWOMBLI.

- It is still unclear whether sometimes "branching" is a real occurrence or merely an apparent one that is seen when two fibers at different depths crisscross in space (at

distinct planes undetected due to the magnification used for image acquisition of the reconstruction of z planes). Authors should discuss this pitfall and provide alternative means for confirmation.

The reviewer raises a valid point similar to reviewer#1 point#5. The typical input images envisaged for analysis are either physical sections or optical sections. These have only a very limited z dimension relative to their x and y dimensions - 4-5 microns for histological sections and 1-2 microns for confocal sections. This means that the issue of two fibres being in different z planes but being interpreted as a branchpoint will be relatively rare. Nonetheless, the reviewer's comment is well made. To examine this issue we have directly compared the number of normalised branchpoints detected if a single confocal section acquired with a narrow pinhole, a single confocal section acquired with a wide pinhole, and z-stack of confocal sections are compared. This is now described in detail in lines 204 - 214 and the metrics are highlighted in green in Table S1.

- Regarding Supplemental Figure 1B: Could the original image(s) that served as input(s) for be shown? Also, can the authors provide an output numerical readout (per mask), like in Figure 3A, so the user could understand how the image quality affects output numbers?

We now show the images in the revised Supplementary Figure 4 (which was previously Supplementary Figure 1).

- Could MATLAB and OrientationJ (or similar popular tools) results be shown as comparison to TWOMBLI using the same one or two image examples? This will help the user understand what exactly is the TWOMBLI output informing on in comparison to available tools. This is important because, after trying out the TWOMBLI plugin, this user ended up questioning how results obtained differ and/or add to results obtained from MATLAB and/or OrientationJ.

TWOMBLI includes end-points, branch points, gap analysis (which is crucial when considering the likelihood of cell migration through an ECM or the size of epithelial structures within a tissue), and fractal dimension (which gives a measure of complexity not related to alignment). The alignment metric from TWOMBLI is the same as the dominant direction metric given by OrientationJ. If a user is more interested in local alignment, then it would be pertinent to use OrientationJ as it can derive local alignment and vector fields. However, the aim of TWOMBLI to characterize the global characteristics of ECM: therefore, we provide a global alignment metric. Related to this, TWOMBLI provides output quantification in a single csv file for the image as a whole, rather than information on individual fibres.

The MATLAB plug-in CurveAlign largely focuses on individual fibres and produces metrics within the TACS framework. TACS1 predominantly reflects increased ECM density, which is reflected in TWOMBLI's HDM metric. TACS2 (alignment of ECM) is reflected in TWOMBLI's alignment metric. However, TACS3 (alignment with respect to

the tumor boundary) is not encapsulated by TWOMBLI. In the event that a user is interested in quantification of ECM with respect to the tumor boundary it would be appropriate to use CurveAlign instead of TWOMBLI.

We are reluctant to present a direct comparison with OrientationJ and CurveAlign as it would not be appropriate to provide detailed commentary on tools generated by others. Without directly engaging with the creators of other tools the potential for inadvertently misrepresenting an aspect of their tool is very real. This would just lead to confusion for the field. However, if the reviewer and editor feel it appropriate then we will include the table below documenting the different TWOMBLI outputs and if there are related outputs generated by OrientationJ and CurveAlign.

Table 1: TWOMBLI metrics and comparison with CurveAlign/OrientationJ

Metric	Reason for inclusion	Inclusion in CurveAlign/OrientationJ
Lacunarity	Complexity	-
Total Length	For normalisation purposes eg computing normalised branchpoints	OrientationJ
Endpoints	Number of fibres	CurveAlign
HGU	Endpoints/TotalLength	-
Branchpoints	Complexity	-
Fractal dimension	ECM Complexity	-
Curvature	Global structural information	CurveAlign
HDM	Reflects the amount of matrix, related to TACS1	CurveAlign
Alignment	Influences cell migration and the mechanical properties of the ECM. Linked to cancer progression and TACS2	CurveAlign/OrientationJ
Gaps	Global structural information	-

- TWOMBLI seems to be more complex (both to use and in the manner outputs are provided) than OrientationJ. Can the authors comment on this and provide advice?

We respectfully disagree that TWOMBLI is more complex to use than OrientationJ. The experience of local users who have tested TWOMBLI is that they are equivalent in their ease of use. It is true that TWOMBLI generates more metrics. This is very deliberate as there is more to ECM pattern than simply fibre orientation. An additional point is that the numerous requests by reviewers for additional features not provided by either OrientationJ or the first version of TWOMBLI has resulted in a slightly 'heavier' TWOMBLI.

- What is the biological meaning of the GAPs and differences in fractal dimension?

Three metrics unique to TWOMBLI as compared to CurveAlign and OrientationJ are lacunarity, fractal dimension and branch points. These metrics are different measures of structural complexity. The complexity of ECM patterns is hard to fully characterize with Euclidean metrics alone. It is as yet unclear if/how these complexity metrics are biologically significant. TWOMBLI is fast and ideally suited to batch processing many images. Many scientists have large libraries of ECM patterns which could shed light on the relevance of ECM complexity. Additionally, the optional gap analysis is a unique feature as compared to CurveAlign and OrientationJ, which aims to give information about the structural positioning of fibres in relation to non-fibre areas, for example regions of adipose tissue or dense cellular regions. This is apparent in the prostate exemplars provided. The glandular areas of normal prostate have a higher lacunarity score as the epithelial glands are devoid of fibrillar collagen. This is now described in line 326. These areas also have a low fractal dimension. In contrast, the stromal areas of the normal prostate have a high fractal dimension indicating the complexity of the matrix network. As expected, the stromal area has a high matrix density score. Alignment is not a particularly useful metric to discriminate the different prostate images. This highlights the importance of the diverse metrics included in TWOMBLI.

Explaining the biological justification for all metrics will greatly assist the user and could increase the potential impact of utilizing this new tool.

We agree and have now amended the text in many places to highlight the biological utility of the metrics.

- In Supplemental Figure3: It appears that samples graphed left to the lacunarity line seemed to cluster with canonical aligned ECMs, can this be noted in the Figure? In other words, could authors mark the isotropic vs. anisotropic results in this graph so the user can start to distinguish what they are used to discern from what is newly available?

The reviewer is correct. The canonical aligned ECMs are to the left of the lacunarity line and lie towards one end of the alignment line (as would be expected). Of note, the MAF2 ECM images lie to the upper end of the lacunarity line compared to most of the other non-aligned ECMs. This reflects the larger gaps in the ECM - visible in Figure 5a - and is an example of the 'new' information that TWOMBLI can provide.

- It will be very useful if the (very helpful) cartoons that were provided in the left panels of Figure 2, or a version similar to these, will be included again for each of the metrics presented in Figures 5B and 5C (and throughout the figures as key).

This is now done as requested in Figure 5.

- Regarding supplementary Figure 4: Could samples representative of "B" color coding (i.e., normal glands (yellow), normal stroma (red), and tumour (blue) based on normalized HDM, fractal dimension, and lacunarity metrics), be noted in "A" so the user could conceptually connect between the two panels?

This has been done. Thank you for the suggestion.

- It could help the user if TWOMBLI was to include graphical outputs (similar to OrientationJ) akin to graphs presented in supplementary Figures 3 and 4.

We are a bit confused by this comment. If the reviewer is requesting graphical outputs of the matrix network that is analysed (shown in Supplementary Figure 4) then these are available in the masks folder. If the reviewer is requesting multi-dimensional analysis of the measurements (such as the PCA in Supplementary Figure 3), then we would argue that this is best done by the user who will have the best understanding of the data, and not generated automatically.

- To emphasize the multi usage capabilities, nerves, angiogenesis, etc. and added example to the provided actin for a gland or sprouting of vessels or nerves could be included. In other words, added emphasis for the ability of multiple usages is sought. For this, the discussion could start with a broad need for quantifying biological patterning and then narrow down to the need for ECM patterning assessments as an example. This way the reader will see the full potential of TWOMBLI (the title may also need to be more inclusive).

We agree with the reviewer that the tool has the potential to be applied to many contexts and we now provide an additional example of its use on endothelial networks in Supplementary Figure 6. This is described in the text in lines 338-343. However, we are reluctant to completely restructure the manuscript as the reviewer proposes. This is largely because each different class of input data (actin, microtubules, nerves, blood vessels etc) is likely to require slightly different optimisations and variations of the output metrics. Doing this comprehensively is a very large task and would require some specialist knowledge of the different biological systems. This is beyond the scope of this manuscript. Our intention is to highlight the potential versatility of the tool is that other researchers with specialist knowledge and different requirements are able to build upon TWOMBLI.

- The discussion and methods portions of this type of reports are perhaps as important, or more valuable, than the results. Hence, authors may want to add text similar to the documentation file as part of the methods (not solely as supplemental material), while a link to the demonstration video should be front and central.

We agree and have now added several additional references to the demonstration video on lines 214 and in the abstract. Regarding the relative split of information between the methods and the documentation file, we have tried to avoid excessive repetition and focused on making the revised main text accessible to a diverse readership, who may not all be interested in some of the more detailed aspects.

- The opening portion of the discussion, regarding ECM patterning being understudied, is inaccurate; ECM architecture has been given much attention ever since the late Dr. Keely vetted that previously observed CDM patterning is also evident in vivo; using SHG (multi-photon microscopy).

We now realise that this sentence did not convey exactly the meaning that we intended. Our intention was not to overlook the amazing work that the reviewer highlights, but to convey that many tumour microenvironment researchers continue to overlook the importance of the ECM and fail to integrate it into their analyses. We have moderated the text so that it more accurately reflects this - lines 349-358.

- Also regarding the discussion: an important aspect of TACs is that these inform on the orientation of the fibers with regards to the orientation of the tumor border (differences between TACs2 and TACs3), could the authors please clearly recognize that TWOMBLI lacks such information/output and propose for it to be integrated in future developments? Similarly, spatial distribution of ECM patterns with regards to other structures, nerves, immune cells, tumor etc., all of which could be recognized from the pathological slides used (i.e., Picosirius red), could also be recognized as caveats or aspects for future implementation. Only then TWOMBLI could truly be a "one stop shop".

The reviewer is correct that TWOMBLI does not generate TACS information and we never claimed that it did. We now make this clear in lines 356 - 358 and explain how the HDM metric relates to TACS1 and the alignment metric relates to TACS2.

Reviewer #3 (Comments to the Authors (Required)):

Authors have developed an image J plugin which enables to analyze patterns of extracellular matrix. Changes in matrix are associated with many diseases and measuring parameters such as matrix density, alignment, curvature etc. of fibers is clinically relevant.

Whilst I think this plugin may be useful for a variety of cell biologists and to some extent clinicians, to my opinion the tool falls short of the criteria set by JCB in that "papers

presenting methods should describe a technological advance of broad/general interest that permits the interrogation of cell biological problems in ways previously impossible ...". The plugin is essentially a program stitched together from existing image J modules and does not represent a technological advance. As mentioned by the authors themselves, there are a number of quite sophisticated algorithms around that can perform similar analysis of matrix as the macro described in this manuscript.

We are pleased to note that the reviewer finds the tool will be useful. As the manuscript is now under consideration for LSA, the comment regarding suitability for JCB is no longer relevant. TWOMBLI has many features not covered in existing tools, such as OrientationJ and CurveAlign, including fractal dimension, lacunarity, and branchpoints.

January 5, 2021

RE: Life Science Alliance Manuscript #LSA-2020-00880-TR

Erik Sahai
Cancer Research UK - London
Cancer Research UK
Centre for Cell & Molecular Biology
Institute of Cancer Research
London, 237 Fulham Road UK-London SW3 6JB
United Kingdom

Dear Dr. Sahai,

Thank you for submitting your revised manuscript entitled "A FIJI Macro for quantifying pattern in extracellular matrix". We would be happy to publish your paper in Life Science Alliance pending final revisions necessary to meet our formatting guidelines.

Along with the points listed below, please also attend to the following:

- please upload both your main and supplementary figures as single files
- please use the [10 author names, et al.] format in your references (i.e. limit the author names to the first 10)
- please add a conflict of interest statement to your main manuscript text
- please add a section with your main and supp. Fig. Legends to your main manuscript text
- please upload your manuscript text as an editable doc file
- please specify the 'Category' of your manuscript at the time of re-submission
- Images from Figure 2 are re-used in Figure 3A and 5C. It is clear that in the latter 2 cases, these images are representatives of the phenotype and not data images, but as per our policy we advise against re-using the same panels. If you have no other option but to re-use them, we request you to clarify in the figure legends that these images have been re-used or duplicated from Figure 2.
- Similarly, images from S2B are also re-used in S3. Please either use a different image or clarify in the legend that the images were re-used
- Would you be willing to change the Manuscript Type to Methods/Tools?

A. FINAL FILES:

B. MANUSCRIPT ORGANIZATION AND FORMATTING:

Sincerely,

Shachi Bhatt, Ph.D.

Executive Editor
Life Science Alliance
<https://www.lsjournal.org/>
Tweet @SciBhatt @LSAJournal

Reviewer #1 (Comments to the Authors (Required)):

I am satisfied with the revised manuscript. I recommend publication.

January 8, 2021

RE: Life Science Alliance Manuscript #LSA-2020-00880-TRR

Erik Sahai
Cancer Research UK - London
Cancer Research UK
Centre for Cell & Molecular Biology
Institute of Cancer Research
London, 237 Fulham Road UK-London SW3 6JB
United Kingdom

Dear Dr. Sahai,

Thank you for submitting your Methods entitled "A FIJI Macro for quantifying pattern in extracellular matrix". It is a pleasure to let you know that your manuscript is now accepted for publication in Life Science Alliance. Congratulations on this interesting work.

DISTRIBUTION OF MATERIALS:

Again, congratulations on a very nice paper. I hope you found the review process to be constructive and are pleased with how the manuscript was handled editorially. We look forward to future exciting submissions from your lab.

Sincerely,

Shachi Bhatt, Ph.D.

Executive Editor

Life Science Alliance

<https://www.lsjournal.org/>
